# Pathogenic Variants in *USH1G*/SANS Alter Protein Interaction with Pre-RNA Processing Factors PRPF6 and PRPF31 of the Spliceosome

**DOI:** 10.3390/ijms242417608

**Published:** 2023-12-18

**Authors:** Jacques S. Fritze, Felizitas F. Stiehler, Uwe Wolfrum

**Affiliations:** Institute of Molecular Physiology, Johannes Gutenberg University Mainz, 55128 Mainz, Germany; jfritze@uni-mainz.de (J.S.F.);

**Keywords:** splicing, U4/U6.U5 tri-snRNP, Usher syndrome, protein–protein interaction, FRET, AlphaFold2, in silico structure predictions

## Abstract

Pre-mRNA splicing is an essential process orchestrated by the spliceosome, a dynamic complex assembled stepwise on pre-mRNA. We have previously identified that *USH1G* protein SANS regulates pre-mRNA splicing by mediating the intranuclear transfer of the spliceosomal U4/U6.U5 tri-snRNP complex. During this process, SANS interacts with the U4/U6 and U5 snRNP-specific proteins PRPF31 and PRPF6 and regulates splicing, which is disturbed by variants of *USH1G*/SANS causative for human Usher syndrome (USH), the most common form of hereditary deaf–blindness. Here, we aim to gain further insights into the molecular interaction of the splicing molecules PRPF31 and PRPF6 to the CENTn domain of SANS using fluorescence resonance energy transfer assays in cells and in silico deep learning-based protein structure predictions. This demonstrates that SANS directly binds via two distinct conserved regions of its CENTn to the two PRPFs. In addition, we provide evidence that these interactions occur sequentially and a conformational change of an intrinsically disordered region to a short α-helix of SANS CENTn2 is triggered by the binding of PRPF6. Furthermore, we find that pathogenic variants of *USH1G*/SANS perturb the binding of SANS to both PRPFs, implying a significance for the *USH1G* pathophysiology.

## 1. Introduction

Pre-mRNA splicing is a fundamental process in eukaryotic cells that almost exclusively occurs in the nucleus. It allows the inclusion/exclusion of exons/introns in the mRNA that can give rise to the synthesis of multiple alternative protein isoforms from a single genomic locus [1]. This process is catalyzed by the spliceosome, a highly dynamic macromolecular complex composed of five small nuclear ribonucleoproteins (U1, U2, U5, U4/U6 snRNPs) and numerous other polypeptides [2]. We have recently shown that the Usher syndrome type 1G (*USH1G*) protein SANS (scaffold protein containing ankyrin repeats and SAM domain) regulates pre-mRNA splicing [3]. Here, SANS mediates the intranuclear transfer of the spliceosomal U4/U6.U5 tri-snRNP complexes from the Cajal body and nuclear speckles, which is essential for the spliceosome activation [3,4]. We found that, during these processes, SANS interacts, among others, with the spliceosomal U4/U6 and U5 snRNP-specific proteins PRPF31 and PRPF6, respectively. *USH1G*/SANS deficiency altered the kinetics of spliceosome activation leading to perturbations in constitutive and alternative splicing of target genes, especially genes related to the human Usher syndrome.

Human Usher syndrome (USH) is a complex autosomal recessive rare genetic disorder (prevalence 1:5000–1:10,000) but represents the most common form of deaf–blindness [5]. To date, at least 11 genes have been classified into four clinical subtypes (*USH1-4*); among them, *USH1* is the most severe subtype with profound hearing loss, vestibular dysfunction, and pre-pubertal onset of retinal dysfunctions in the form of retinitis pigmentosa [5,6,7].

The human *USH1G* gene encodes SANS, a scaffold protein of ~52 kDa molecular weight consisting of 461 amino acids. SANS is composed of three N-terminal ankyrin repeats (ANK1–3), two central domains (CENTn and CENTc), a sterile alpha motif (SAM), and PDZ binding motif at the very C-terminal end (Figure 1) [8,9]. In the auditory hair cells, SANS is essential for the correct arrangement of the stereocilia in the hair bundles during development and for the composition and function of the mechanosensitive tip-link complex at the tip of the stereocilia [10,11,12]. Less is known about the function of SANS in the retina. Previous work indicated that SANS associates with cellular modules such as intracellular transport, endocytosis, and primary ciliogenesis [9,13,14,15,16,17]. Additionally, the role of SANS and other USH proteins in stabilizing the outer segment of photoreceptors has been hypothesized due to their presence in the calyceal processes, which are microvilli-like processes of the inner segment of photoreceptor cells [18]. However, *USH1G*/SANS is not primarily expressed in photoreceptor cells but in Müller glial cells of the retina, as recently highlighted by single-cell RNAseq data [19].

In any case, the integrative scaffold functions of SANS are essential for the proper operation of diverse cellular modules, relying on context-dependent interactions with multiple protein partners. Other teams and ours have identified numerous proteins that bind to all domains and motifs of the SANS molecule [9,13,15,17,20,21,22]. These studies also showed that the CENTn and CENTc domains are the preferential binding sites in SANS, with at least 50 different proteins binding to it [3,9,14,17,20]. Nevertheless, the mechanisms by which the CENTn and CENTc domains of SANS facilitate their interaction with multiple binding partners remain unexplained.

Here, we aimed to gain further insights into the molecular interaction of binding partners with the CENTn domain and to understand the effects of SANS-mediated splicing regulation. We investigated in detail the binary interaction of SANS with both PRPF31 and PRPF6 in cells using FRET assays and in silico deep learning-based protein structure predictions such as AlphaFold2-multimer and Metapredict [23,24]. Our data suggest that the SANS CENTn domain binds directly via two distinct conserved regions (CENTn1, CENTn2) to PRPF31 and PRPF6. We also provide evidence that these interactions occur sequentially and that PRPF6 binding introduces a conformational change of an intrinsically disordered region to an α-helix in SANS CENTn2. Finally, we found that perturbations in the binary interactions between the two PRPFs and pathogenic variants of *USH1G*/SANS were disturbed, implying a significance in the pathogenesis of *USH1G*.

## 2. Results

### 2.1. Binding of SANS to PRPF31 and PRPF6 in the Nucleus Revealed by FRET

We have previously demonstrated the interaction of the *USH1G* protein SANS with the splicing molecules PRPF31 and PRPF6 of the tri-snRNP complex by in vitro pull-down assays [3]. Here, we aimed to explore their binary binding in more detail by fluorescence resonance energy transfer (FRET) acceptor photobleaching assays in the cell [25]. For this, we co-expressed eYFP-SANS together with eCFP-tagged eCFP-PRPF31 or PRPF6-eCFP, in HEK293T cells. Confocal fluorescence microscopy showed that eYFP-SANS co-localized with both PRPF6-eCFP and eCFP-PRPF31, respectively, in the nucleus, as indicated by fluorescence intensity plots and confirmed by positive Pearson coefficient R values (Figure 2A,B). In addition, eCFP-tagged harmonin, a USH-related scaffold protein known to co-localize in situ and to interact with SANS [8,20,26], co-localized with co-expressed eYFP-SANS as expected (Figure 2C). Intriguingly, the co-localization of eYFP-SANS with harmonin-eCFP was predominantly in the cytoplasm. The triple transfection of PRPF6-eCFP, eYFP-SANS, and mRFP-PRPF31 showed co-localization of all three proteins inside the nucleus (Figure 2D), which was consistent with the localization of endogenous proteins as previously reported [3].

To study the binary interaction of SANS with both PRPFs in the cell, we performed FRET acceptor photobleaching assays in the nucleus (Figure 3A). FRET analyses revealed the interaction of the eCFP-eYFP FRET pairs SANS-PRPF31 and SANS-PRPF6 in nuclei (Figure 3B). The obtained FRET signals were at the same height as for the interaction between eYFP-SANS and the known SANS binding partner protein harmonin-eCFP in the cytoplasm (Figure 3B). In addition, we analyzed PRPF3, another PRPF of the spliceosomal tri-snRNP complex, previously shown not to interact with SANS as a negative control [3]. We did not observe notable FRET signals in assays with the SANS-PRPF3 eCFP-eYFP FRET pair (Figure 3C) and in all additional negative controls applied (Appendix A). Taken together, the FRET-based interacting assays revealed specific binding of SANS to PRPF31 and PRPF6, respectively, in the nucleus.

During sequential spliceosome activation, the interaction of PRPF31 and PRPF6 as bridging factors between the small nuclear ribonucleoproteins snRNPs, U4/U6, and U5 of the tri-snRNP complex is essential [27]. This prompted us to test whether SANS interferes with the interaction between PRPF31 and PRPF6 within the tri-snRNP complex. As expected, we found significant FRET signals in HEK293T cells co-transfected with eCFP-PRPF31 and PRPF6-eYFP (Figure 3D). Additionally, expressed mRFP-SANS or mRFP alone did not alter the FRET signals for eCFP-PRPF31 and PRPF6-eYFP. Next, we conducted FRET analysis for eCFP-eYFP FRET pairs of SANS-PRPF31 in the presence of PRPF6-mCherry (Figure 3E). The FRET signals between eYFP-SANS and eCFP-PRPF31 were unaffected by PRPF6-mCherry (Figure 3E). However, co-expression of mRFP-PRPF31 significantly reduced FRET signals between SANS-PRPF6 FRET pairs, which was not observed in the presence of mRFP (Figure 3F). However, FRET efficiency decreased only by approximately 16%, which still indicates the interaction of SANS and PRPF6, but with lower affinity than in the absence of PRPF31. This decrease may indicate the competition of binding partners and/or a change in conformation of the tertiary complex of PRPF31-PRPF6 and SANS.

Taken together, these data suggest that the binary binding of SANS to PRPF31 is independent of PRPF6, but that PRPF31 interferes with the interaction of SANS with PRPF6 (Figure 3G).

### 2.2. PRPF31 and PRPF6 Interact with Different Regions in SANS CENTn Domain

Our FRET experiments indicated that SANS interacts differently with PRPF31 and PRPF6. To explore the structural basis for this interaction, we used AlphaFold2-multimer as an in silico approach for protein–protein complex predictions [23]. The predicted structures are documented on GitHub (https://github.com/LabWolfrum/Fritze_et_al.2023.git). The AlphaFold2 structural predictions aligned to the structures of PRPF31 and PRPF6 previously obtained by cryoelectron microscopy ([28], PDB: 6QW6) (Appendix A). For SANS, AlphaFold2 predicted a central unstructured intrinsically disordered region (IDR) flanked by an α-helix representing the N-terminus of CENTn and the SAM domain in the C-terminus (Appendix A).

To analyze the protein complexes, we used AlphaFold2-multimer [23], which predicts an alignment error (PAE) representing the accuracy of the predicted models, summarized by the PAE_mean_ score for each model (Appendix A). We compared our prediction workflow with the harmonin–SANS complex previously determined by X-ray crystallography as a benchmark [26]. AlphaFold2-multimer predicted a complex similar to the experimentally resolved structure, with a PAE_mean_ ≥ 12.24 Å in the interaction region, namely, SANS-SAM/PBM and the N-terminal harmonin homology domain of harmonin (Appendix A) [29]. It should be noted that during development, AlphaFold2 was trained on accessible PDB data [23,30], and the data for SANS/harmonin structures may have been among them.

AlphaFold2-multimer predicted low confidence (PAE_mean_ ≥ 25.66 Å) for a complex of full-length PRPF31 and SANS, except for two small regions containing the N-terminal domains of both proteins (Appendix A), namely, the N-terminal part of the CENTn domain (aa 128–173) of SANS and the NOP domain (aa 215–333) of PRPF31 (Figure 1). Subsequent AlphaFold2-multimer predictions of these areas predicted a complex of SANS CENTn and the PRPF31 NOP domain, with high confidence of a PAE_mean_ ≥ 11.83 Å (Figure 4A). No other regions of SANS, such as the ankyrin repeats, CENTc domain, and SAM/PBM domain, were predicted in a complex with the PRPF31 NOP domain with such high confidence (Appendix A). The amino acid sequence analyses of these domains predicted that the SANS CENTn-PRPF31 NOP complex is stabilized by multiple hydrophobic interactions (aa 140–165, Figure 4A and Appendix A). In summary, these data suggest a protein complex of SANS N-terminal CENTn domain and PRPF31 NOP domain, stabilized by hydrophobic interactions.

AlphaFold2-multimer predictions for a complex of full-length PRPF6 and SANS were also of low confidence in the model (PAE_mean_ ≥ 31.02 Å) (Appendix A). However, AlphaFold2-multimer predicted a complex of the C-terminal part of PRPF6, containing HAT domains, and a short region of SANS CENTn (aa 166–194) domain, with high confidence of a PAE_mean_ ≥ 10.21 Å (Figure 4B). Interestingly, AlphaFold2-multimer predicted that a short α-Helix (aa 183–198) was formed in this complex by a part of the unstructured region of SANS CENTn domain. No other region of SANS could be predicted with PRPF6 with such high confidence (Appendix A). Taken together, our in silico experiments suggest two different regions of the CENTn domain, hereafter referred to as CENTn1 (aa 128–173) and CENTn2 (aa 174–243). These data suggested direct binding of PRPF31 and PRPF6 to SANS. PRPF31 interacted exclusively with CENTn1, while PRPF6 interacted predominantly with CENTn2 and seven residues of CENTn1, indicating that the binding site for both PRPFs do not overlap (Figure 4C).

To predict the integration of SANS with the tri-snRNP complex, we made use of the AlphaFold2-multimer predictions for SANS structural complexes with PRPF31 and PRPF6 described above (Figure 4A,B) and fitted these in the experimental confirmed tri-snRNP complex ([28], PDB: 6QW6) (Figure 4D,E). However, SANS did not fit into the structure of the tri-snRNP complex in a single conformation based on the predicted interactions with both PRPFs. Also, it should be noted that PRPF31 interaction with SANS (Figure 4D) may clash with SNU13 and the U4-snRNA in the U4.U6 di-snRNP. Nevertheless, in both predicted conformations, SANS was structurally separated from PRPF3. Taken together, these data suggest a spatial separation of SANS binding platform in the tri-snRNP complex.

### 2.3. In Silico Analysis Predicts Evolutionary Conserved Multi-Conformational Intrinsically Disordered Regions (IRDs) for SANS

To decipher the structural characteristics of SANS domains, we used the in silico tool Metapredict [24] in combination with the pLDDT scores of AlphaFold2 (Figure 5A). Metapredict is a bidirectional recurrent neural network that gives a predicted disordered value for each residue. While high values in Metapredict indicate intrinsically disordered regions (IDRs), the pLDDT scores of AlphaFold2 decrease in IDRs [31].

Metapredict and AlphaFold2 predicted the ankyrin repeats (aa 1–126), the CENTn1 (aa 128–177), and the SAM domain (aa 386–447) as folded and three unfolded IDRs in SANS, namely, the CENTn2 domain (aa 178–243), the CENTc domain (aa 278–385), and the PBM (aa 451–461) (Figure 1 and Figure 5A). These predictions also strengthen the observation that the SANS CENTn domain consists of two different regions: a structured region (CENTn1, aa 128–177) and an IDR (CENTn2, aa 178–243).

To analyze the evolutionary conservation of SANS domains, we performed multiple sequence alignments. The alignment of SANS amino acid sequences of more than 98 mammals and 266 vertebrate species demonstrated that SANS overall is highly conserved among species (Figure 5B; Appendix A). Even IDRs, which are commonly known for evolutionary low conservation [32], namely, the CENTn2, CENTc, and the C-terminal PBM, were highly conserved overall among vertebrate and mammal species (Figure 5B and Appendix A).

Next, we analyzed the physicochemical properties of the SANS CENTn2 domain by applying the CIDER analyzer [33]. The CENTn2 of SANS was predicted by CIDER to be a Janus sequence (Figure 5C). Janus sequences can change their structure depending on the binding partner [34]. Furthermore, CIDER predicted that SANS CENTn2 is positively charged overall (Appendix A), a feature that is known to foster protein binding [35].

In summary, our results indicate three distinct conserved IDRs in CENTn2, CENTc, and PBM of SANS. Moreover, the physicochemical properties predicted for SANS CENTn2 indicate a high potential for protein–protein interactions, which is confirmed by the binding of PRPF6 to this domain described above.

**Figure 5 ijms-24-17608-f005:**
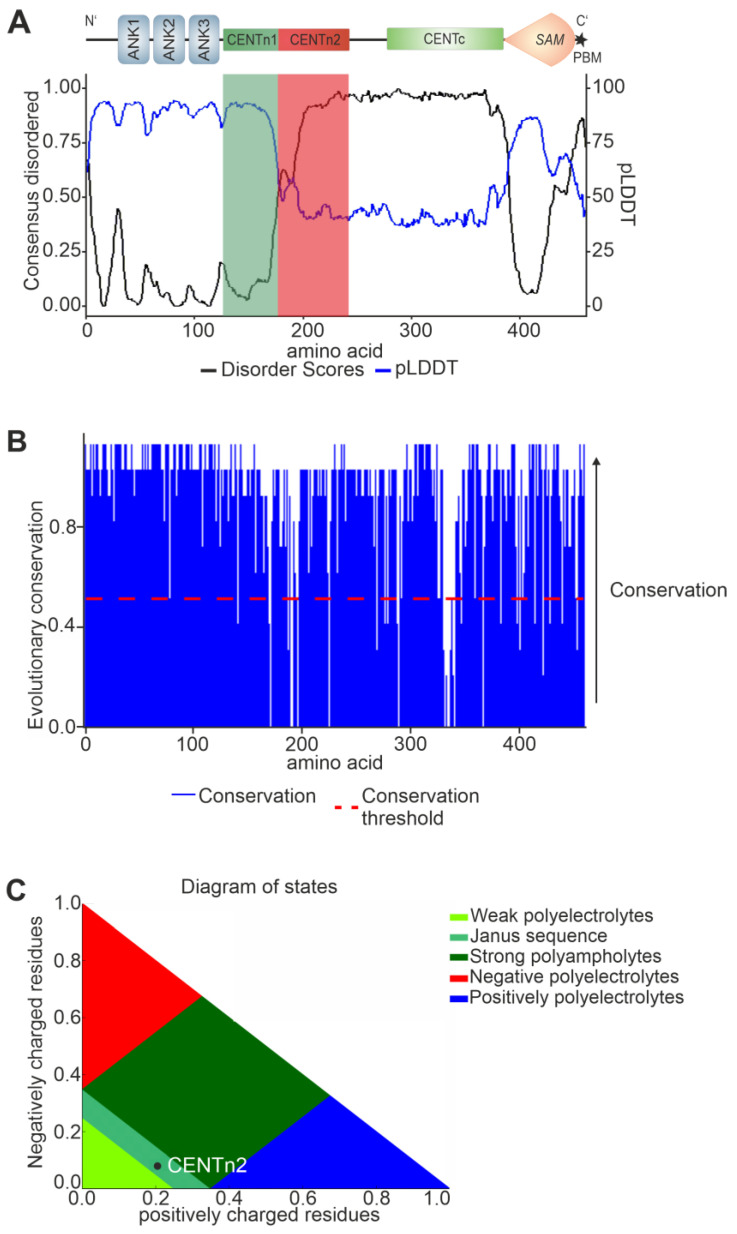
Predictions of SANS domain structure and their evolutionary conservation. (**A**) SANS predicted disordered state by Metapredict (black line) and AlphaFold2 (blue line). An increase of Metapredict prediction (disordered score) above 0.5 or a decrease in the AlphaFold2 prediction (shown by pLDDT) below 50 indicates an intrinsically disordered region (IDR). Metapredict and AlphaFold2 predicted SANS CENTn1 (green box) as structured and SANS CENTn2 (red box) as IDR. (**B**) Evolutionary conservation (blue) of SANS proteins for 266 vertebrates. The conservation threshold (red dotted line) indicates high conservation of SANS in vertebrates, except for small regions of CENTn and CENTc domains. (**C**) SANS CENTn2 (black dot) in a Das–Pappu plot predicted by CIDER is identified as a flexible Janus sequence.

### 2.4. USH-Causing Pathogenic Variants of SANS Show Altered Interaction with PRPFs

We have previously shown that pathogenic variants in the *USH1G* gene interrupt the interactions in the ternary USH protein complex of SANS, *USH2A*, and whirlin [9]. Here, we tested whether *USH1G* pathogenic frameshift mutations *USH1G*/SANS^V132Gfs*3^ and *USH1G*/SANS^S278Pfs*71^ (www.LOVD.nl/USH1G, accessed on 25 July 2023), which lead to a premature stop in the CENTn1 or CENTc domain, respectively (Figure 1), also alter the binary interaction between SANS and PRPFs.

We co-expressed eYFP-tagged versions of both *USH1G*/SANS mutations, with either eCFP-tagged PRPF31, PRFP6, or harmonin as control, in HEK293T cells (Appendix A). Confocal microscopy and fluorescent intensity plots demonstrated that both *USH1G*/SANS variants still co-localized with both PRPFs in the nucleus, confirmed by the positive Pearson coefficients (R) (Appendix A). However, while in co-expression with the *USH1G*/SANS variants, many PRPF6 droplets were still distributed all over the nucleus, as in the *USH1G*/SANS wildtype expressing cells, where fluorescent PRPF31 appeared in a few larger droplets in the nucleus. The expression of *USH1G*/SANS variants destroyed the specific localization of harmonin in the cytoplasm and led to an even distribution of the proteins throughout the cell (Appendix A).

Subsequent FRET assays with eYFP-SANS^S278Pfs*71^ and both eCFP-tagged PRPFs showed significant reductions in FRET efficiencies compared to eYFP-SANS (Figure 6A,B). This indicated that the S278Pfs*71 mutation disrupts the binding of SANS to PRPF31 and PRFP6. However, we predicted no change in the IDR for SANS^S278Pfs*71^ compared to SANS (Appendix A).

In FRET assays with eYFP-SANS^V132Gfs*3^ and PRPF6-eCFP, FRET efficiencies were also significantly reduced compared to the efficiency of eYFP-tagged full-length SANS and PRPF6-eCFP (Figure 6C).

Interestingly, in FRET assays with eYFP-SANS^V132Gfs*3^ and eCFP-PRPF31, FRET efficiency was not reduced, indicating a binding of PRPF31 also to the N-terminal part of SANS (Figure 6D). Since this finding was contradictory to our previous results from in vitro pull-down assays [3], we validated this observation by additional FRET assays with the truncated eYFP-SANS-Nterm construct (Figure 1) and eCFP-PRPF31 (Figure 6E). eCFP-eYFP FRET pairs of SANS-Nterm-PRPF31 showed no reduction of FRET efficiency compared to eYFP-tagged full-length SANS supporting binding of PRPF31 to the N-terminal part of SANS, which mainly consists of three ankyrin repeats (Figure 1).

Both *USH1G*/SANS variants (V132Gfs*3 and S278Pfs*71) did not show a significant FRET signal in all FRET controls performed (Appendix A). In addition, FRET assays with harmonin, which are known to bind to the SAM/PBM region of the C-terminus of SANS, which is absent in both *USH1G*/SANS variants, showed significantly reduced efficiencies compared to full-length eYFP-SANS (Appendix A). Thereby, interaction with both fluorescent-tag proteins, eYFP and eCFP, could be excluded. Moreover, we confirmed the correct expression of all eYFP-SANS constructs via Western blot analysis (Appendix A). Further, AlphaFold2-multimer did not predict a model for PRPF31 full-length or coiled-coil domain and SANS-Nterm with high confidence (Appendix A).

Taken together, our data showed that both pathogenic variants, *USH1G*/SANS^V132Gfs*3^ and *USH1G*/SANS^S278Pfs*71^, disrupted the interaction with PRPF6. In contrast, the interaction of PRPF31 was disrupted for *USH1G*/SANS^S278Pfs*71^, but not with the shorter eYFP-SANS^V132Gfs*3^ revealing a binding site for PRPF31 in the N-terminal part of SANS alternative to the binding to SANS CENTn1.

## 3. Discussion

Pre-mRNA splicing is a fundamental process in eukaryotic cells that almost exclusively occurs in the nucleus, catalyzed by the spliceosome, a highly dynamic macromolecular complex composed of small nuclear ribonucleoproteins (snRNPs) and numerous other polypeptides [1]. We previously identified the *USH1G* protein SANS as one of these proteins regulating pre-mRNA splicing by mediating the intranuclear transfer of the spliceosomal U4/U6.U5 tri-snRNP complexes from the Cajal body and nuclear speckles essential for the spliceosome activation [3]. During these processes, SANS interacts through the CENTn domain with the spliceosomal U4/U6 and U5 snRNP-specific proteins PRPF31 and PRPF6, respectively [3]. However, questions of the mode of binding of SANS to PRPFs, by direct or mediated by other spliceosome proteins, or even by RNA, remained unanswered. In addition, the exact binding site in the CENTn domain of the SANS protein, which comprises more than one-fourth of the entire protein, has not yet been determined.

In our study, we found FRET efficiencies for both eCFP-eYFP FRET pairs, PRPF31-SANS and PRPF6-SANS, in the range of the tandem eCFP-c-eYFP construct (positive control), were significantly higher than the negative controls. Since positive FRET signals are only achieved at a distance of 8–10 nm or less between the two complex partners [25,36], these findings not only confirm the interactions of PRPF31 and PRPF6 to SANS, but also suggest that both PRPFs bind directly to SANS without any mediator involved as previously considered [37]. Microscopic analyses, including fluorescence intensity plots and consistently positive Pearson coefficient R values, demonstrated the co-localization of PRPFs with SANS in the cell nucleus, further supporting a direct binary interaction.

Using AlphaFold2-multimer, we pinpointed the binding of PRPF31 and PRPF6 to different, separate regions of the CENTn domain of SANS, namely, to aa 140–165 of the long α-helix of CENTn1 and to aa 166–194, respectively. The predicted binding of PRPF31 via the NOP domain to the helical structures of SANS was also found for the interaction of PRPF31 with the helical structures in the small nuclear ribonucleoprotein 13 (SNU13) (also known as 15.5k protein or NHP2-like protein 1) of the U4/U6.U5 tri-snRNP complex [28]. Both observations are in line with the common interaction modes of NOP domains present in other ribonucleoproteins [38].

For the interaction of PRPF6 with SANS, we confirmed C-terminal HAT domains as the interacting region in PRPF6 previously reported [3]. These domains bind to the seven amino acids of the α-helix of SANS CENTn1 but mainly to a region of SANS CENTn2, which was consistently predicted as an evolutionary conserved IDR by Metapredict and AlphaFold2. Here, we note that the model of the PRPF6-SANS complex had low overall confidence by AlphaFold2-multimer, probably since AlphaFold2-multimer is trained on structured protein data [23]. Nevertheless, the model is supported by the ability of PRPF6 to bind IDRs in other proteins, such as the translation initiation factor eIF4E enrolled in splicing [39].

Our in silico predictions by AlphaFold2-multimer also suggested that this IDR of the CENTn2 forms a newly derived small α-helix upon binding to PRPF6. This structural change is supported by the observation that SANS CENTn2 represents a highly conserved Janus sequence, a protein region with the characteristic to specifically change the structure by binding different proteins [34]. The properties of a Janus sequence in the SANS CENTn domain are probably also the basis for allowing the interaction of the numerous additional binding partners that have been previously identified [3,9,14,21].

Although our data consistently indicate that the binding of SANS to PRPF31 and PRPF6 is mediated by distinct subdomains of the SANS molecule, we observed a decrease in FRET efficiency for the PRPF6-SANS pair upon co-transfection with PRPF31, indicating a competition between the two PRPFs. This finding is in line with the fact that the complexes predicted for SANS and the two PRPFs did not simultaneously fit into the structure of the U4/U6.U5 tri-snRNP complex [28]. These results suggest that SANS molecules bind to both PRPFs separately, either sequentially in the tri-snRNP complex or independently to the snRNP subunits. Sequential binding of proteins to the tri-snRNP complex was previously shown for U6-LSm ring proteins, which also change their spatial arrangement in the tri-snRNP complex, binding first to PRPF6 and then to PRPF3 during the formation of spliceosomal complex B [40]. Nevertheless, separate binding of SANS to the two PRPFs could also occur during the formation of the tri-snRNP complex in the Cajal bodies when the U4/U6 snRNP containing PRPF31 and U5 snRNP harboring PRPF6 maturate separately from each other [41]. Alternatively, in the tri-snRNP complex, SANS may preferentially bind to PRPF31, as indicated by our FRET data, and interacts with PRPF6 during the recycling process of the tri-snRNP complex components after activation of the spliceosome when sufficient free U5 snRNP/PRPF6 is present [2,42]. Although the latter alternative is supported by previous work showing that SANS deficiency does not affect the maturation of the tri-snRNP complex but does affect the recycling of U5-snRNP after activation of the spliceosome [3], in the tri-snRNP complex, the SANS binding site of PRPF31 might already be occupied by SNU13 and U4-snRNA, which could interfere with the binding of SANS [28].

More than 70 pathogenic variants in *USH1G*/SANS have been identified thus far in *USH1* patients ([43]; www.LOVD.nl/USH1G, accessed on 25 July 2023). However, the molecular mechanisms underlying the pathophysiology caused by these defects remain poorly understood. Our previous study provided evidence that SANS is part of the pre-mRNA splicing machinery and demonstrated that pathogenic variants in *USH1G*/SANS, including both frameshift mutants, *USH1G*/SANS^V132Gfs*3^ and *USH1G*/SANS^S278Pfs*71^, investigated here, alter the pre-mRNA splicing of target genes [3]. These genes include *USH1C*, which is frequently alternatively spliced in the human retina [3,44]. Present FRET assays showed that for both pathogenic variants, the binding of PRPF6 was disrupted. *USH1G*/SANS^S278Pfs*71^ did not interact with either PRPF6 or PRPF31 in the cell, although the binding sites for both PRPFs in the CENTn domain are still present in the truncated SANS variant. Neither Metapredict nor AlphaFold2 predicted structural changes in *USH1G*/SANS^S278Pfs*71^ that could interfere with the binding of the two PRPFs. However, the content of proline residues in the frameshift sequence changes from 2 to 9 in *USH1G*/SANS^S278Pfs*71^, which might lead to steric changes in the CENTn and CENTc domains and the binding sites for the PRPFs interfering with the binding of PRPF6 and PRPF31.

The short, truncated SANS variants, SANS^V132Gfs*3^ and the SANS-Nterm construct, lack the entire CENTn1 domain, which we identified above as the binding site of PRPF31, and all other downstream domains. However, both truncated SANS molecules consistently interact with PRPF31 in present FRET assays in cells, which was previously not found in in vitro pull-down assays [3]. The interaction of PRPF31 and SANS^V132Gfs*3^ is in line with the co-localization of both polypeptides in the nucleus of the cell observed by our microscopy analyses. Consequently, this novel finding suggests a binding site for PRPF31 in the three ankyrin repeats of the N-terminal portion of SANS alternative to SANS CENTn1. Ankyrin repeats are common protein–protein interaction platforms [45], and we have recently demonstrated that a set of intraflagellar transport (IFT) molecules bind to the ankyrin repeats of SANS [17]. Unfortunately, we failed to predict a site for PRPF31 that can bind to the region of ankyrin repeats in the N-terminus of SANS using the prediction tools employed.

In recent years, pathogenic variants leading to retinitis pigmentosa (RP) have been associated with many genes encoding proteins of the U4/U6.U5 tri-snRNP complex [4,46,47]. Besides PRPF31 (RP11) and PRPF6 (RP60), this list includes several other components, such as PRPF8 (RP13), PRPF3 (RP18), SNRNP200 (RP33), and PRPF4 (RP70). Here, we showed that mutations in *USH1G*/SANS lead to disruption of binary binding properties with the key molecules of the U4/U6.U5 tri-snRNP complex, PRPF31 and PRPF6. Vice versa, several pathogenic RP mutations in the genes of the two PRPFs are also located in binding sites to SANS ([48]; https://www.hgmd.cf.ac.uk/ac/index.php, accessed on 27 July 2023) and most likely also result in alterations in their interaction with SANS. Perturbations in the molecular interactions between the components of the U4/U6.U5 tri-snRNP complex are consistent with the fact that mutations in PRPF6, PRPF31, and *USH1G*/SANS have almost identical effects on splicing and that similar pathomechanisms leading to RP also cause similar retinal phenotypes.

Alternative splicing certainly increases the diversity of the transcriptome and proteome. Compared to other tissues, the retina exhibits the highest rates of alternative splicing [49,50]. Specific splicing programs are essential for the function and maintenance of the retina, which is uniquely regulated in a highly complex manner [51,52]. Thus, alterations in the splicing machinery may affect the retina more prominently than other tissues, which may explain why pathogenic variants in *USH1G* and in the genes of other splicing factors predominantly lead to retinal defects.

## 4. Materials and Methods

### 4.1. DNA Constructs and Primers

All nucleic acids used are listed in Appendix A.

### 4.2. Cloning

cDNAs for expression of proteins were obtained by RT-PCR from human HEK293T cells and cloned into pENTR^TM^, as described by the manufacturer (Thermo Fisher, Karlsruhe, Germany, #K240020). Gateway^TM^ LR reaction into the appropriate destination vector was performed according to the manufacturer’s protocol (Thermo Fisher, #11791020). USH-causing mutants and SANS-Nterm, PRPF31, and PRPF6 were cloned into pENTR^TM^, with the template described previously [3]. eCFP-c-eYFP was cloned from the eYFP vector into the pDest vector with eCFP.

### 4.3. Cell Culture and Cell Lines

Dulbecco’s modified Eagle’s medium (DMEM #31966021) containing 10% heat-inactivated fetal calf serum (FCS) (Thermo Fisher #A4766801 or Cytiva, Freiburg, Germany, #SV30160.03) was used to culture HEK293T cells (ATCC: CRL-3216). Cells were fixed with 4% PFA in PBS for 10 min at room temperature. Nuclear DNA was stained by SPY650-DNA (TebuBio, Offenbach, Germany, #SC501).

### 4.4. Fluorescence Co-Localization Observation

HEK293T cells were seeded in 24 well plates with a density of 40,000 cells/well. Cells were transfected with Lipofectamine^TM^ LTX, as described by the manufacturer (Thermo Fisher #15338100). Fixed HEK293T cells were observed with a Leica TCS SP5. Images were analyzed with Fiji 64 v5 [53]. The Pearson coefficient was performed with the Fiji Plugin *Coloc 2* for the whole nucleus. The intensity plot was performed over the indicated region of interest with the Fiji Plugin *Plot Profile*.

### 4.5. Fluorescence Resonance Energy Transfer (FRET) Acceptor Photobleaching Assay

HEK293T cells were seeded in six well plates with a density of 500,000 cells/well. Cells were transfected with Lipofectamine^TM^ LTX, as described by the manufacturer (Thermo Fisher #15338100). Fixed HEK293T cells were analyzed with a Leica TCS SP8 and FRET acceptor photobleaching was performed following the Leica protocol (https://downloads.leica-microsystems.com/TCS%20SP8/Application%20Note/FRET_AB_with%20SP8-AppLetter_EN.pdf, accessed on 1 January 2023). eCFP was used as the donor (D) and eYFP as the acceptor (A). The acceptor was bleached at 100% laser intensity for 10 repeats. FRET efficiency was calculated via: ${FRET}_{eff}=\frac{{(D}_{post}-D_{pre})}{D_{post}}$. To exclude cellular structure, the mean of six regions of interest (ROI) in the bleached area was calculated and FRET efficiency of an unbleached region (≥3 µm away from bleached ROI) was subtracted from the mean. Bleach efficiency was calculated via the formula ${Bleach}_{eff}=\left(1-\frac{A_{post}}{A_{pre}}\right)\;*\;100$. Only ROIs with >60% bleach efficiency were used for analysis. Data were normalized by the positive control eCFP-c-eYFP if indicated. Example images of FRET acceptor bleaching are shown in Appendix A.

### 4.6. AlphaFold2-Multimer

AlphaFold2-multimer prediction was performed using the hetero-oligomer options of ColabFold [54]. The FASTA sequence of the respective protein was used from UniProt (https://www.uniprot.org, accessed on January–April 2023) and a colon between the respective sequences simulated complexes. The following sequences were used: SANS: Q495M9; PRPF31: Q8WWY3; PRPF6: O94906; harmonin: Q9Y6N9. AlphaFold2 notebook was used in the ColabFold version (https://colab.research.google.com/github/sokrypton/ColabFold/blob/main/AlphaFold2.ipynb#scrollTo=G4yBrceuFbf3, accessed on 12 July 2023) and the standard options were extended by the template mode pdb70/pdb100, model type AlphaFold2-multimer_v2 recycling 24 times. Predicted structures are documented in GitHub (https://github.com/LabWolfrum/Fritze_et_al.2023.git). Structures were visualized, inspected, and superimposed using PyMOL (https://pymol.org/2, accessed on 1 August 2022, version: 0.99rc6) or UCSF Chimera V5 [55], which was also used to generate all structure images. PAE_mean_ was calculated as the mean of all PAE in the corresponding PAE plot area. Contacts for some protein complexes (Appendix A) were predicted by the UCSF Chimera tool “Find Clashes/Contacts” (https://www.cgl.ucsf.edu/chimera/docs/ContributedSoftware/findclash/findclash.html, accessed on 3 March 2023). The following options were chosen: themselves, contact (default criteria −0.4). As output, an overlap and a distance value can be observed.

The predicted protein complexes were fitted to a previously published PDB structure of the human tri-snRNP complex ([28], PDB: 6QW6).

### 4.7. Evolutionary Conservation

A previously described protocol was used as a foundation [56]. The complete code can be found on GitHub (https://github.com/LabWolfrum/Fritze_et_al.2023.git). The evolutionary conservation was performed with the help of Jupyter Notebook and Jalview [57]. Python code was designed with the help of ChatGPT-3.5 (OpenAI). For graphs, areas not present in human SANS were excluded.

### 4.8. Western Blot

HEK293T cells were seeded in six well dishes with a confluency of 500,000 cells per/well. Cells were transfected with Lipofectamine^TM^ LTX, as described by the manufacturer (Thermo Fisher #15338100). Cells were lysed 24 h after transfection with TritonX-100, protein amount was measured with a BCA assay, and proteins were eluted with 5x Laemmli buffer, separated by SDS-PAGE followed by Western blotting. Monoclonal mouse-anti-GFP (cross-reaction to eYFP, Proteintech, Planegg-Martinsried, Germany # 66002-1-Ig) was used.

### 4.9. Additional Bioinformatic Analyses

CIDER was accessed via the online platform (http://pappulab.wustl.edu/CIDER/, accessed on 2 February 2023) and the FASTA sequence was analyzed with default options. SANS USH-causing mutants were chosen through the online Usher syndrome database (www.LOVD.nl/USH1G, accessed on 25 July 2023). Metapredict was used by the Python library option [24]. PDB structures were obtained from the PDBe webserver (https://www.ebi.ac.uk/pdbe/, accessed in March 2023).

### 4.10. Statistics

Statistical analysis was performed with R-Studio version 4.3.2 [58]. The statistical methods and the significance criteria are listed in corresponding individual legends. Results are shown as a boxplot of data from at least three separate experiments. Significance was determined as: * *p* ≤ 0.05, ** *p* ≤ 0.009, *** *p* ≤ 0.0009.

## 5. Conclusions

Our data strengthen the evidence that SANS participates in alternative splicing by interaction with components of the U4/U6.U5 tri-snRNP complex. For this SANS directly binds with different binding sites of its CENTn to the splicing factors PRPF31 and PRPF6. Our data also suggest that SANS molecules bind sequentially to the two PRPFs in the tri-snRNP complex (Figure 7). Finally, we provide evidence that perturbations in the molecular interaction of SANS with both PRPF molecules underlie the pathophysiology of pathogenic variants in *USH1G*/SANS, leading to the ocular phenotype in USH1.

## Figures and Tables

**Figure 1 ijms-24-17608-f001:**
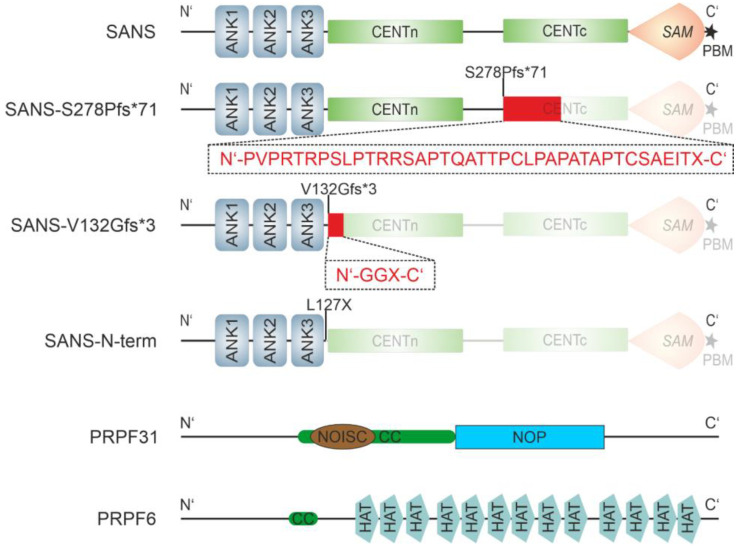
Domain structures of SANS variants and the pre-mRNA processing factors (PRPF) PRPF31 and PRPF6. Domain structure of SANS and USH-causing pathogenic variants SANS^S278Pfs*71^, SANS^V132Gfs*3^, SANS-Nterm, PRPF31, and PRPF6. SANS consists of three ankyrin repeats (ANK1–3), a central domain (CENT), divided into the N-terminal CENTn and the C-terminal CENTc, a sterile alpha motif (SAM), and PDZ binding motif (PBM, asterisk). Both USH-causing mutants, S278Pfs*71 and V132Gfs*3, are frameshift mutations, which lead to “missense extension” (red boxes and amino acid sequences below) and a premature stop in CENTc (S278Pfs*71) or CENTn (V132Gfs*), respectively. SANS-Nterm is an artificial construct missing CENTn and onward. PRPF31 consists of coiled-coil (CC, green) regions, a NOISC domain (brown), named after the central domain in Nop56/SIK1-like proteins, and a NOP domain (blue), a ribonucleoprotein (RNP) binding module, with both found in various pre-RNA processing ribonucleoproteins (InterPro: http://www.ebi.ac.uk/interpro/entry/InterPro/IPR012976 and IPR036070, accessed on 1 July 2023). PRPF6 exhibits thirteen half TPR (HAT) domains, as well as one CC domain.

**Figure 2 ijms-24-17608-f002:**
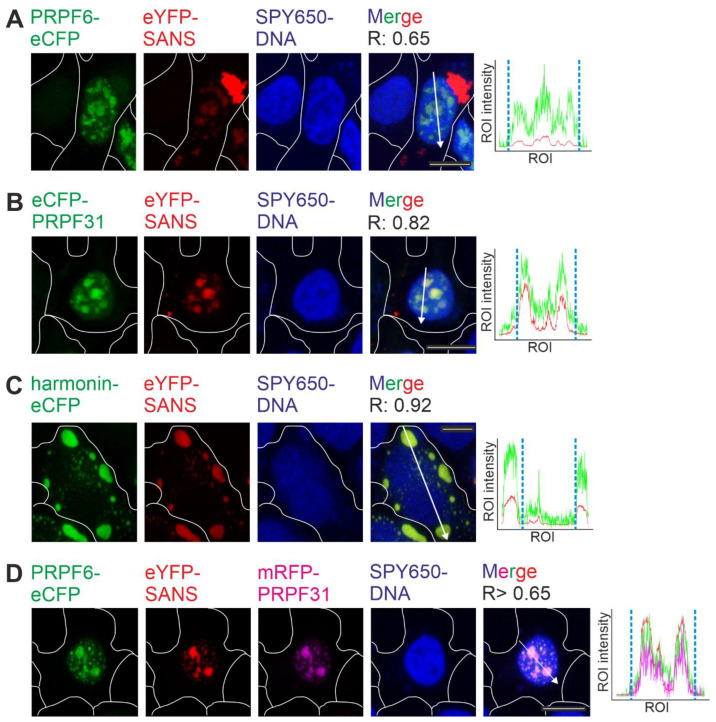
Co-localization of SANS and splicing proteins PRPF6 and PRPF31. (**A**–**D**) Confocal microscopy analyses of HEK293T cells co-transfected with eYFP-SANS and eCFP/mRFP tagged proteins, counterstained with SPY650-DNA as a nuclear marker. Fluorescence intensity plots of eYFP SANS (red) and transfected eCFP- (green) and mRFP-tagged (magenta) proteins for regions of interest (ROI) indicated by white arrows in merge images; blue dashed lines indicate nuclear extension. Positive Pearson coefficient R values indicate co-localization. (**A**) PRPF6-eCFP co-localized with eYFP-SANS in the nucleus. (**B**) eCFP-PRPF31 co-localized with eYFP-SANS in the nucleus. (**C**) *USH1C* protein harmonin-eCFP co-localized with eYFP-SANS, predominantly in the cytoplasm outside of the nucleus. (**D**) eYFP-SANS co-localized with PRPF6-eCFP and mRFP-PRPF31 in the nucleus of HEK293T cell in a triple transfection. White lines indicate cell borders. Example images of *n* = 3 experiments; scale bar = 10 µm.

**Figure 3 ijms-24-17608-f003:**
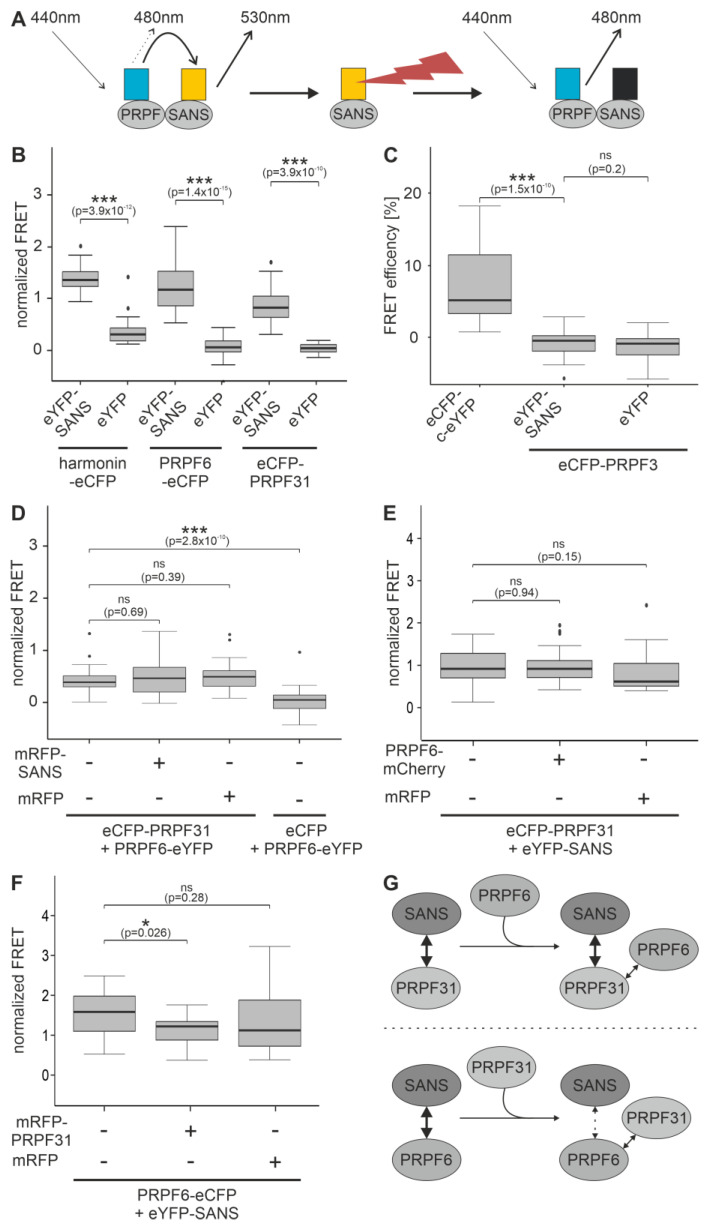
Interaction of SANS and binding partners analyzed by FRET acceptor bleach in HEK293T cells. (**A**) Illustration of the principle of the FRET acceptor bleach assay. Interaction of two proteins tagged with either eCFP (blue) or eYFP (yellow) leads to FRET (left). The acceptor (eYFP) is bleached (red flash symbol) (middle), which leads to increased emission of the donor (eCFP) (right) and no fluorescence of acceptor (black). (**B**–**F**) FRET assay in co-transfected HEK293T cells. In (**B**,**D**–**F**), FRET efficiencies were normalized to the fused eCFP-c-eYFP FRET pair (positive control), as shown in (**C**) and Appendix A. (**B**) FRET pairs eYFP-SANS-harmonin-eCFP, -PRPF6-eCFP, and -eCFP-PRPF31, respectively, show a significant increase in the normalized FRET efficiencies compared to eYFP FRET pair negative controls. (**C**) FRET efficiency of eYFP-SANS-eCFP-PRPF3 shows a FRET efficiency similar to the eYFP negative control, which is significantly different from the positive control eCFP-c-eYFP. (**D**) FRET efficiencies of the eCFP-PRPF31-PRPF6-eYFP pair in the absence and presence of mRFP-SANS do not significantly differ. (**E**) Normalized FRET efficiencies of the eCFP-PRPF31-eYFP-SANS in the absence and presence of PRPF6-mCherry do not significantly differ. (**F**) Normalized FRET efficiency of the PRPF6-eCFP-eYFP-SANS is significantly decreased in the presence of mRFP-PRPF31. In (**E**,**F**), mRFP was used as a control for the third interaction partner. Outliers are shown as dots above/below the boxplots. Wilcoxon signed-rank test was performed for three independent experiments. (**G**) Schematic interaction of SANS, PRPF31, and PRPF6 from (**D**) to (**F**). SANS interaction with PRPF6 is decreased in the presence of PRPF31.

**Figure 4 ijms-24-17608-f004:**
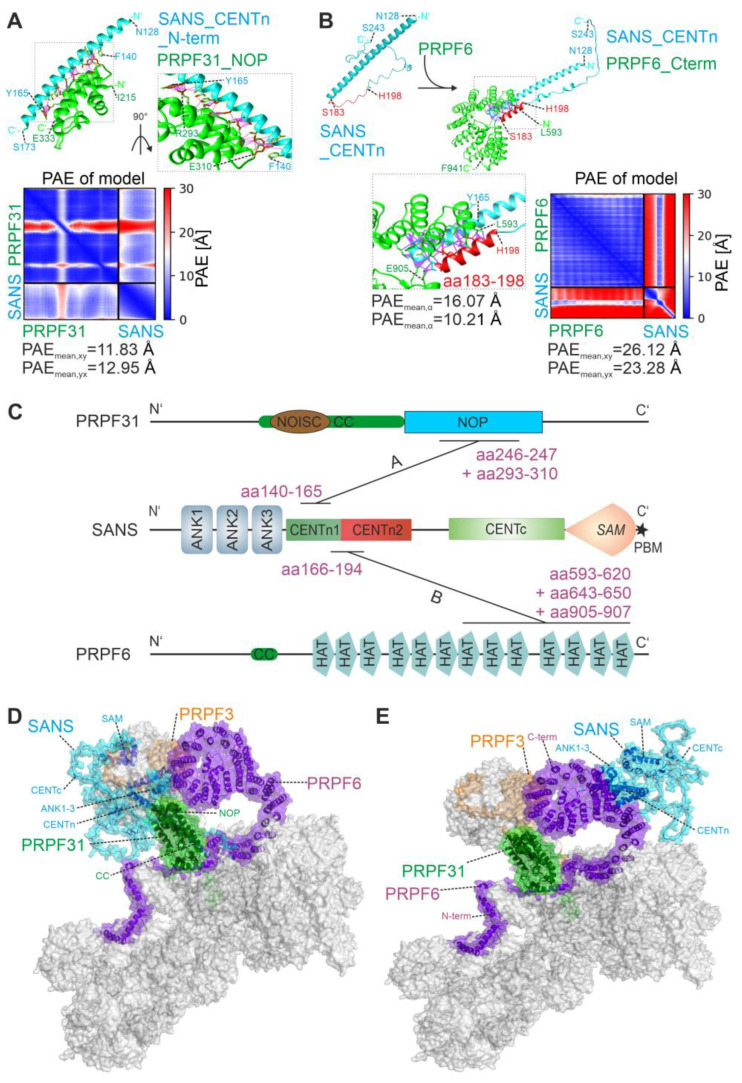
Molecular structure predictions of the interaction of SANS with PRPF31 and PRPF6. (**A**,**B**) In silico predictions of SANS and PRPFs complexes by AlphaFold2. Purple lines indicate hydrophobic interactions (Appendix A). (**A**) PRPF31 NOP domain (green, amino acids (aa) 215–333) binds to the N-terminal part of SANS CENTn (blue, aa 128–173). Close-up is shown on the right (zoomed box). (**B**) PRPF6 binds via aa 593–941 of the C-terminus (green) to the SANS CENTn domain (blue, aa 128–243); PAE is predicted with high confidence for a small region of the two proteins. In SANS CENTn, a small α-Helix (red, aa 183–198) is formed from an unstructured region upon interaction with PRPF6 (zoomed box). (**C**) Illustration of the binding regions on the domain structure of SANS, PRPF31, and PRPF6 predicted (**A**,**B**); aa of binding sites are indicated in purple. Note: SANS CENTn is divided into CENTn1 (aa 128–173) and CENTn2 (aa 174–243) based on structural differences (see Figure 5A,B). (**D**,**E**) Illustration of SANS binding to the U5.U4/U6-tri-snRNP complex (cryoEM structure, [28], PDB: 6QW6), as predicted in (**A**,**B**). (**D**) SANS (blue) fits into the tri-snRNP complex in a small pocket between PRPF31 (green) and PRPF6 (purple) when interacting with PRPF31. (**E**) SANS position in the tri-snRNP complex changes when interacting with PRPF6.

**Figure 6 ijms-24-17608-f006:**
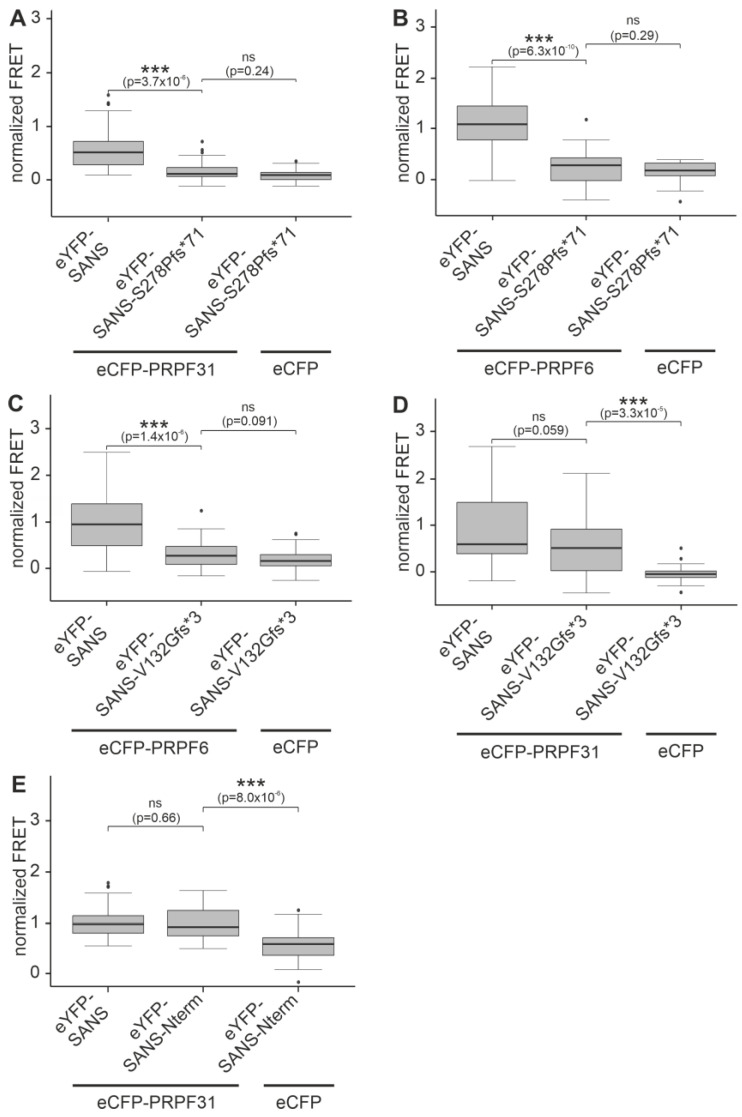
Interaction of pathogenic variants of *USH1G*/SANS to PRPF31 and PRPF6 analyzed by FRET in HEK293T cells. (**A**,**B**) FRET assays with eYFP-SANS, eYFP-SANS^S278Pfs*71^ paired with eCFP-PRPF31 (**A**), and PRPF6-eCFP (**B**) normalized to FRET of eCFP-c-eYFP fused tandem pair. Normalized FRET efficiencies are significantly decreased in both SANS^S278Pfs*71^-PRPF pairs to levels of the negative control eYFP-SANS^S278Pfs*71^-eCFP compared to eYFP-SANS, indicating no binding of eYFP-SANS^S278Pfs*71^ to both PRPFs. (**C**) Normalized FRET efficiency is also decreased in the eYFP-SANS^V132Gfs*3^-PRPF6-eCFP pair, indicating no binding of SANS^V132Gfs*3^ to PRPF6. (**D**,**E**) Normalized FRET efficiencies of eYFP-SANS^V132Gfs*3^ (**D**) and eYFP-SANS-Nterm (**E**) paired with eCFP-PRPF31 are similar to eYFP-SANS-eCFP-PRPF31 pair values, but significantly different from eCFP control pairs, indicating no binding of both truncated variants to PRPF31. Outliers are shown as dots above/below the boxplots. Wilcoxon signed-rank test was performed for three independent experiments.

**Figure 7 ijms-24-17608-f007:**
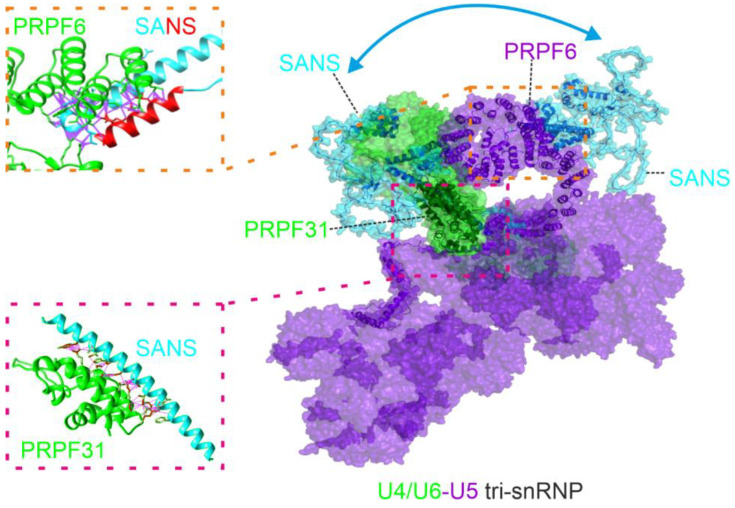
Schema of proposed conformation of SANS in the tri-snRNP complex. SANS interacts with PRPF31 and PRPF6 in the tri-snRNP. Two conformations of SANS are used to bind to either PRPF6 or PRPF31, respectively.

## Data Availability

All raw data are available through contacting the corresponding author. Python code and AlphaFold2-multimer predictions used in this study are available via GitHub (https://github.com/LabWolfrum/Fritze_et_al.2023.git).

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
