# Peer review of "Pathogenic Variants in USH1G/SANS Alter Protein Interaction with Pre-RNA Processing Factors PRPF6 and PRPF31 of the Spliceosome"

_ijms, 2023, doi:10.3390/ijms242417608_

Round 1

Reviewer 1 Report

Comments and Suggestions for Authors

Wolfrum et al. offer a detailed understanding of the molecular interaction between pre-RNA processing factors PRPF6 and PRPF31 of the spliceosome, and the USH1G protein SANS. Their findings suggest that SANS molecules sequentially bind to both PRPFs, as determined through the use of FRET assays and computational structure prediction tools. Furthermore, they propose that disruptions in the molecular interaction between SANS and both PRPF molecules contribute to the pathophysiology of certain pathogenic variants. While the research appeared technically sound, I still have two major concerns.

1. The findings derived from the AlphaFold2-multimer predicted structures necessitate experimental confirmation. For instance, the predicted structure implies that SANS binds to the two PRPFs via two distinct conserved regions of its CENTn. This could be further substantiated through mutagenesis experiments of crucial interacting residues observed in the structure.

2. Further studies are necessary to examine the short α-helix of SANS CENTn2, which is triggered by the binding of PRPF6, as indicated by the predicted structure. At least, a comprehensive molecular dynamics simulation trajectory could be conducted to understand the dynamics and stability of the short α-helix.

Reviewer 2 Report

Comments and Suggestions for Authors

1. I am confused by the use of 'CENT' instead of 'CENTn' in lines 64, 65, and 431..

2. I suggest the authors consider using ChimeraX to build contact-network-based topological domains and color the structure accordingly (Tools/Structure Prediction/AlphaFold Error Plot/Color PAE Domains). This feature could provide a more solid basis for domain selection and would significantly enhance the figures and narrative in the results section, especially where protein-protein interactions are detailed.

3. I would reconsider the use of the PAEsum as  confidence proxy. Would it not be more appropriate to use the median value in the region? Furthermore, on which basis authors define low and high confidence threshold values?

4. Regarding the hydrophobic amino acid substitution virtual experiment: I advise caution in silico mutagenesis with AlphaFold2, particularly without examining/reporting the conservation of the involved residues. If the authors choose to proceed, they should provide a strong rationale for the changes and consider exploring other hydrophobic amino acid combinations. Given that this experiment does not underpin subsequent evaluations or hypotheses, it might be more prudent to omit it entirely.

Round 2

Reviewer 1 Report

Comments and Suggestions for Authors

This revised manuscript can be accepted for publication.